# Spatial and temporal distribution of ribosomes in single cells reveals aging differences between old and new daughters of *Escherichia coli*

**Lin Chao\*[†], Chun Kuen Chan[†], Chao Shi, Ulla Camilla Rang**

Department of Ecology, Behavior and Evolution, School of Biological Sciences, University of California San Diego, La Jolla, United States

## eLife assessment

This study is a potentially **important** contribution to the field of protein biosynthesis pathways and their link to aging, especially regarding the thorough analysis of variation in measures expected to correlate with elongation rate in old and new daughter cells derived from old and new mother cells. However, the imaging results, analysis, and methodologies are **incomplete**, as in its current form several key questions remain unanswered.

**\*For correspondence:**
lchao@ucsd.edu

[†]These authors contributed equally to this work

**Competing interest:** The authors declare that no competing interests exist.

**Abstract** Lineages of rod-shaped bacteria such as *Escherichia coli* exhibit a temporal decline in elongation rate in a manner comparable to cellular or biological aging. The effect results from the production of asymmetrical daughters, one with a lower elongation rate, by the division of a mother cell. The slower daughter compared to the faster daughter, denoted respectively as the old and new daughters, has more aggregates of damaged proteins and fewer expressed gene products. We have examined further the degree of asymmetry by measuring the density of ribosomes between old and new daughters and between their poles. We found that ribosomes were denser in the new daughter and also in the new pole of the daughters. These ribosome patterns match the ones we previously found for expressed gene products. This outcome suggests that the asymmetry is not likely to result from properties unique to the gene expressed in our previous study, but rather from a more fundamental upstream process affecting the distribution of ribosomal abundance. Because damage aggregates and ribosomes are both more abundant at the poles of *E. coli* cells, we suggest that competition for space between the two could explain the reduced ribosomal density in old daughters. Using published values for aggregate sizes and the relationship between ribosomal number and elongation rates, we show that the aggregate volumes could in principle displace quantitatively the amount of ribosomes needed to reduce the elongation rate of the old daughters.

## Introduction

A mother *Escherichia coli* cell divides asymmetrically into two daughter cells that have different elongation rates (*Stewart et al., 2005*; *Chao, 2010*; *Shi et al., 2020*; *Proenca et al., 2018*; *Proenca et al., 2019*; *Łapińska et al., 2019*). The new daughter elongates faster while the old daughter is slower. The asymmetry and old-new designation of the daughters result from the fact that the rod-shaped mother divides by forming new poles at its midplane. As a result, all *E. coli* cells are polarized at birth with a new and an old pole (*Figure 1A*). When a polarized cell in turn becomes a mother and divides, its daughters will also be polarized but they can also be identified as new

**Figure 1.** Cell division and polarity in *E. coli*. (**A**) Assignment of old (red) and new (blue) poles and daughters. The starting cell is white because it was randomly picked to start the experiment and its polarity was unknown. Because the division plane (dashed line) cuts the white cell at the midpoint of the long axis, the poles formed at the division point are new and the distal poles are old. At the next division, the new daughter acquires the new pole, while the old daughter receives the old pole. Note that two divisions must be tracked to determine the old and new daughters from the starting white cell. The outlines of the bottom four daughters in the figure are colored red and blue to identify them as old and new daughters, while the intracellular red and blue colors identify the old and new poles, also designated as O and N. Although old and new poles and daughters are here tracked for only two generations, the notation and tracking methods can be extended into generations 3, 4, 5, 6, 7, and further if needed. For example, if the old and new poles of any of the four daughters after two divisions are known as in (**A**), and these daughters elongate, become a mother, and divide to produce two grand-daughters, the polarity of the grand-daughters can be determined by the same tracking methods. (**B**) Time-lapse images of an *E. coli* bacterium dividing into two and four cells. Top row: phase contrast. Second row: assignment of old (red) and new (blue) poles from the top row cells. Third row: fluorescence images matching phase images in time and position. Bottom row: heatmap of fluorescence images reporting ribosome density in the top row cells, showing lesser intensity by the old poles (blue color spots) than the new poles and inside the cells (purple color). Scale on the right shows intensity gradient. A plot of ribosomal density versus length as a continuous variable is provided in *Figure 1—figure supplement 1*. This supplement plot is presented only for visualization and does not represent the format used in the data analyses to follow. (**C**) Schematic showing division of a cell into two halves containing either the new pole or the old pole. The ratio of the fluorescence in the new half divided by that in the old half (or pole ratio) was used to normalize and pool different old and new daughter pairs, when each pair descended from a different mother. (**D**) Schematic showing division of a cell into four length quartiles denoted NP, L2, L3, and OP in relation to the old and new poles. These quartiles were used to quantify the distribution of fluorescence along the length of a cell.

The online version of this article includes the following figure supplement(s) for figure 1:

**Figure supplement 1.** Fluorescence profile along new and old daughter pairs' cell lengths at birth.

and old daughters depending on whether they acquired the mother's new or old pole. Note that mothers can also be denoted as new or old, depending on whether they were new or old daughters at birth. The difference in elongation rate between old and new daughters was found by *Lindner et al., 2008* to correlate negatively with the amount of damaged proteins (inclusion bodies) in individual cells. They also found that protein aggregates were more often associated with the old pole of the mother, in which case damaged proteins are partitioned asymmetrically by the mother to its daughters. However, a study by *Govers et al., 2018* has reported a weaker and nonsignificant negative correlation. Because the two studies used different external stressors (to induce damage) and fluorescent reporters, they are not fully comparable and the relationship between elongation rate and the damaged proteins in *E. coli* remains unresolved (see 'Discussion' for additional details). Despite the differences between Lindner et al. and Govers et al., protein aggregates often have a negative qualitative effect on cell health and viability (*Maisonneuve et al., 2008*; *Gatti-Lafranconi et al., 2011*; *Schramm et al., 2020*; *Rang et al., 2018*), and more so in old daughters (*Vedel et al., 2016*; *Winkler et al., 2010*).

The asymmetry of both the elongation rate difference between old and new daughters and the damage partitioning by the mother in *E. coli* has evolutionary consequences. The asymmetry creates fitness variance between daughters that increases the effectiveness of natural selection (*Chao, 2010*; *Chao et al., 2016*), and an asymmetrical lineage has more progeny over time relative to a symmetrical one. An analogy comes from economic models. Two asymmetrical $500 bank accounts at 10 and 6% yield more money over time than two symmetrical $500 accounts at 8%. However, the asymmetry comes with costs. While the higher elongation rate of the new daughter rises in the lineage over generations, the rate of the old daughter declines (*Stewart et al., 2005*; *Chao, 2010*). Under more benign culture conditions, the rise and the decline can achieve stable values and the lineages are immortal (*Chao, 2010*; *Proenca et al., 2018*; *Łapińska et al., 2019*; *Rang et al., 2011*; *Rang et al., 2012*). Under more stressful and damage-inducing conditions, the old daughter lineage can die because its elongation rate declines to zero and the death probability is dosage dependent (*Proenca et al., 2018*; *Proenca et al., 2019*). The bacterial population survives despite the death of the old daughter lineage because the new daughter lineage is still able to achieve a stable rate that is greater than zero (*Chao, 2010*; *Proenca et al., 2018*; *Proenca et al., 2019*). Because the elongation rate declines and death of the old daughter lineage is a form of physiological aging, bacteria can be used to investigate the evolutionary origins of biological aging. Despite a quantitative correlation between the amount of damaged proteins and elongation rates remaining unresolved (see above), the asymmetries here described for elongation rates and death strongly suggest that damage stress in *E. coli* may play a key, if not causal, role in cell aging. We note that the asymmetrical partitioning of damage is not an alternative to protein repair, or vice versa. Protein repair (including reprocessing) is occurring (*Ezraty et al., 2017*; *Merdanovic et al., 2011*) and its effect is to lower the damage rate experienced by the cell. However, if damage aggregates are present in cells, the damage rate must necessarily exceed the repair rate. There is likely a cost to repair, in which case the repair rate may become limited by diminishing returns. Thus, the asymmetry could be an evolutionary adaptation for dealing with a low repair rate. If repair rates were sufficiently high to negate damage, *E. coli* should not have evolved asymmetry.

To develop further our understanding bacterial aging, our previous study examined the distribution of expressed native (not fused to another protein) GFP molecules in *E. coli* (*Shi et al., 2020*). We found that the GFP density was higher in new poles than in old ones and in cells with a faster elongation rate. To take the analysis one step further, we present here results from our examination of the distribution of ribosomes in the poles and daughters of single *E. coli* cells. We used the *yfp* (yellow fluorescent protein) and S2 subunit (*rpsB* gene) fusion by *Bakshi et al., 2012* as a reporter for ribosomal density. The S2-YFP construct is an effective fluorescent reporter. The S2 ribosomal subunit makes a desirable reporting partner because free subunits are negligible and nearly all cellular S2-YFP is incorporated into ribosomes (*Bakshi et al., 2012*; *Sanamrad et al., 2014*). Almost all of the detected fluorescent S2-YFP comes from copies are already incorporated in ribosomes (*Bakshi et al., 2012*). *Bakshi et al., 2012* used S2-YFP to successfully make a more detailed map of the nucleoid/ribosome segregation. Most importantly, because fluorescent reporters can produce artifacts (*Govers et al., 2018*; *Swulius and Jensen, 2012*; *Landgraf et al., 2012*), YFP and S2-YFP were extensively tested to validate that the reported fluorescence was not artifactual. These tests are detailed next.

The tests validating YFP and S2-YFP followed the methods of *Landgraf et al., 2012* by which validation requires that either fluorescent aggregation, clusters, or foci are reported when non-uniformity is expected by an assay or alternatively not reported when uniformity is expected. An artifact occurs when uniformity is reported when non-uniformity is expected (and vice versa). Because reporting artifacts depend on the protein to which a fluorescent reporter is fused (*Landgraf et al., 2012*), a reporter that makes artifacts with one protein does not necessarily make it with a second. Thus, many of the tests were targeted specifically to the S2-YFP construct. (i) A first test focused on native YFP (not fused to any protein) because an earlier study suggested that YFP produced artifactual non-uniformity due to self-aggregation (*Govers et al., 2018*). This test was needed because the self-aggregation results were based on a fusion construct of the chaperone IbpA and YFP. However, was the self-aggregation a property of YFP or an interaction with IbpA? *Bakshi et al., 2012* found that native YFP reported a uniform distribution, in which case YFP is not inherently prone to self-aggregation artifacts (see also 'IbpA-YFP self-aggregation' in 'Discussion'). (ii) If YFP is fused to OmpR, which is a regulator that binds to the OmpC and OmpF loci, and the three loci (OmpR-YFP, OmpC, and OmpF) are expressed from the bacterial chromosome (hereafter strain A), the fluorescence distribution is uniform (*Batchelor and Goulian, 2006*). Because the chromosome has a low copy number, most OmpR-YFP is distributed over the entire cell. Some OmpR-YFP is bound to OmpC and OmpF but they are too few to produce a fluorescence focus. If OmpC and OmpF are cloned into the multi-copy plasmid pBR322, which forms clusters in cells (*Pogliano et al., 2001*), and the plasmid is inserted into strain A (making strain B), a fluorescence focus reporting the plasmid cluster emerges (*Batchelor and Goulian, 2006*). Additionally, if native OmpR (not fused to YFP or any fluorescent reporter) is over-expressed from a second plasmid that is inserted into strain B (making strain C), the fluorescence focus disappears because the more numerous native OmpR outcompetes OmpR-YFP in binding to the pBR322/OmpC-OmpF cluster. Thus, OmpR-YFP is responding correctly when either uniformity or non-uniformity is expected. (iii) To confirm that the nucleoid/ribosome segregation they observed was not due to YFP reporting artifacts, Bakshi et al. repeated their experiments by adding rifampicin and chloramphenicol, two antibiotics that affect the distribution of the nucleoid DNA (*Cabrera et al., 2009*). Whereas the nucleoid is expanded and rendered more uniform by rifampicin, it is compacted and clumped by chloramphenicol. *Bakshi et al., 2012* found that the fluorescence reported by S2-YFP became more uniform with rifampicin but more clumped and non-uniform with chloramphenicol. As the nucleoid expanded with rifampicin, the ribosome distribution also expanded as it filled the spaces created by the nucleoid expansion. On the other hand, as the nucleoid contracted with chloramphenicol, the ribosomes were excluded from the space occupied by the more clumped nucleoid and they in turn became less uniform. In summary, these YFP, OmpR-, and S2- controls give us confidence that S2-YFP is a reliable reporter for ribosomes and that reported fluorescence foci are better explained by the non-uniform properties of the cell rather than any inherent property or artifact of YFP. We find these tests that functionally validate S2-YFP to be more convincing than examining S2 fused with more fluorescent reporters. If two reporters disagree, which one is the artifact? If they agree, are they both truthful or artifactual?

Our examination of the distribution of ribosomes with S2-YFP could be particularly instructive. First, while Bakshi et al. have shown that ribosomes congregate in the poles, it is not known if ribosomal densities differ between old and new daughters or their poles. Second, our previous report of more expressed gene products in new poles and daughters (*Shi et al., 2020*) could have resulted from regulatory factors acting on a specific gene or at a higher upstream level such as ribosomal abundance. In the latter case, the consequences would be much more fundamental as they could affect the entire proteome. The fact that ribosomal density and elongation rates are highly correlated in bacteria (*Schaechter et al., 1958*; *Neidhardt and Magasanik, 1960*; *Kjeldgaard and Kurland, 1963*; *Poulsen et al., 1995*; *Greulich et al., 2015*) additionally reinforces the importance of knowing the distribution of ribosomes in the poles and daughters of *E. coli*. Lastly, given that ribosomes occupy nearly 40% of the cytoplasmic volume of an *E. coli* cell (*Woldringh and Nanninga, 1985*) and that both ribosomes and damage aggregates tend to reside in the poles, the possibility that the two could be excluding each other at the poles becomes a probability. However, a space-limiting model as a hypothesis for explaining bacterial aging requires first that the distribution of ribosomes in old and new poles or daughters be characterized. Here, we present the results of our study quantifying the temporal and spatial distribution of ribosomes in single *E. coli* cells by time-lapse microscopy.

# Results

Following 'Materials and methods', the distribution of ribosomes was measured in the *E. coli* strain AFS55 with a fluorescent *yfp* reporter fused to the ribosomal subunit S25 (*Bakshi et al., 2012*). Fluorescence levels were quantified from time-lapse images of single cells cultured on agarose pads. Because the level recorded for a cell on an image is affected by the light that the cell loses by diffraction and gains by diffraction from neighboring cells, the diffractional scatter of all images was corrected by deconvolution. Red and blue colors in the figures represent old and new daughters or mothers, respectively. Ratio data (*Figure 1C*) were derived from estimates that were not normalized. All other data for cell length, ribosomal density, and time were normalized when pooling and comparing cells. All error bars are ± SEM and statistical details are presented in figure legends.

We first examined the density of ribosomes in colonies starting from a single cell. A quick visual inspection of the fluorescent images and their heatmaps for a few representative cells showed clearly that the ribosomal densities tended to be highest near the poles (*Figure 1B*). However, high densities also built up in the mid-region of a mother cell right before division. As the mid-region leads to the formation of the new poles of the daughters, the buildup explains how those poles acquired their higher levels. These polar concentrations of ribosomal densities matched the previous ones reported by *Bakshi et al., 2012*. To obtain a more quantitative analysis of the changes in ribosomal densities during cell growth, we collected a larger sample size and compared measurements between old and new mothers, old and new daughters, between their poles, along the length of their cells, and during their elongation from birth to division.

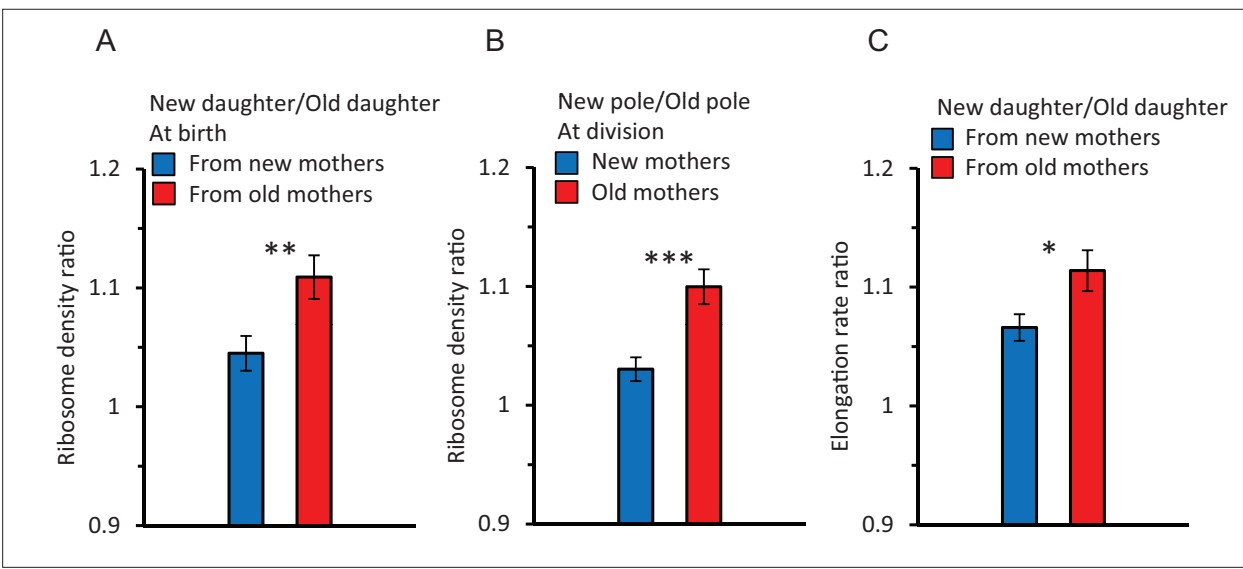

**Figure 2.** Ratios of ribosome density and elongation rates within and between cells. Because ratios are a form of normalization, their values were derived from non-normalized data. Asterisks in figures indicate levels of significance from comparison between ratios of new (blue bar) versus old (red bar) mothers (*, **, *** significance at p<0.05, 0.01, and 0.001). Error bars are SEM. Values of p in the text below are for the significance probability that a ratio is greater than 1.0, unless indicated otherwise. Sample size for ratios is for pairs of data: n = 89 pairs correspond to 89 old/new daughters of the same mother. Each pair is used to obtain one ratio to yield 89 ratios. (**A**) Ribosome ratio of daughters (new/old) at birth from old (red) and new (blue) mother cells. The ratio was 1.11 ± 0.018 from old mothers (p=6.89 × 10⁻⁶, one-tailed paired *t*-test, n = 89 pairs) and 1.04 ± 0.015 from new mothers (p=0.04, one-tailed paired *t*-test, n = 91 pairs). The two ratios were also significantly different from each other (p=0.007, two-tailed non-paired *t*-test, ** in figure). (**B**) Ribosome ratio (new/old) of the two polar halves (*Figure 1C*) from old (red) and new (blue) mothers at division. The ratio in old mothers was 1.10 ± 0.015 (p=6.23 × 10⁻⁵, one-tailed paired *t*-test, n = 89 pairs) and from new mothers 1.03 ± 0.010 (p=0.08, one-tailed paired *t*-test, n = 91 pairs). The difference between the two ratios was significant (p=1.3 × 10⁻⁴, two-tailed non-paired *t*-test, *** in figure). A comparison of the daughter and polar half ratios from old mothers (**A** and **B**; red bars) found no significant difference (p=0.62; two-tailed paired *t*-test). A likewise comparison for new mothers (**A** and **B**; blue bars) also found no significance (p=0.27). (**C**) Elongation rate ratio of new over old daughters from new (blue) and old (red) mothers. The ratio from old mothers was 1.11 ± 0.017 (p=1.3 × 10⁻¹⁰, one-tailed paired *t*-test, n = 216 pairs) and from new mothers 1.07 ± 0.011 (p=2.76 × 10⁻⁷, one-tailed paired *t*-test, n = 198 pairs). The two ratios were significantly different from each other (p=0.02, two-tailed non-paired *t*-test; * in figure). Note that these elongation rate ratios parallel the ribosomal pattern of a higher asymmetry in daughters from old mothers (**A**).

## New daughters at birth from old mothers have more ribosomes

A first comparison revealed that if a pair of old and new daughters descending from the same mother were compared, ribosome density measured by the fluorescence was higher for new daughters. The ratio of reported fluorescence between new/old daughters (the daughter ratio) was 1.08 ± 0.012, which was significantly greater than 1.00 (p=4.38 × 10⁻⁶). However, if the mothers were split into pools of old and new mothers (*Figure 2A*), the daughter ratio from old mothers was 1.11 ± 0.018 and significantly greater than 1.00 (p=6.89 × 10–6) and the ratio from new mothers was 1.04 ± 0.015 and barely significant (p=0.04). Thus, the higher ribosome density in new daughters resulted primarily from old mothers. The difference between old and new mothers was further demonstrated by a strong significant difference between their daughter ratios (1.11 vs. 1.04; p=0.007). Thus, new daughters have more ribosomes relative to old daughters produced by the same mother, but asymmetry is higher in pairs from old mothers. This higher density of ribosomes in new daughters correlates qualitatively with our previously reported higher levels of expressed gene products and elongation rates in new daughters (*Chao, 2010*; *Shi et al., 2020*; *Proenca et al., 2018*; *Proenca et al., 2019*).

## Ribosomal asymmetry between daughters is spatially in place in mothers before division

To further explore our finding of a ribosomal asymmetry between new and old daughters, we examined the intracellular distribution of ribosomal fluorescence in the new and old pole halves (see diagram in *Figure 1C*) of old and new mothers right before division. The two pole halves were transformed into a ratio (new/old) for analysis (*Figure 2B*). We found that pole ratio in old mothers was significantly greater than 1.00 (1.10 ± 0.015; p=6.23 × 10⁻⁵). The pole ratio in new mothers was much lower (1.03 ± 0.010) and not significantly greater than 1.00 (p=0.08). The pole ratios between new and old mothers were significantly different from each other (p=1.3 × 10⁻⁴). Thus, the density of ribosomes in the old and new poles of mothers matches well the density in the old and new daughters of both old and new mothers (*Figure 2A and B*). This match argues that the physical placement of ribosomes in the mother at the time of division sets in place the ribosome distribution in the daughters. To test this argument, we compared directly the ribosomal pole ratio of an individual mother against the ribosomal daughter ratio of the same mother. No significant difference was found, in both old and new mothers, when the ribosomal ratio of the poles and daughters was compared. With new mothers, the pole and daughter ratios were respectively 1.03 vs. 1.04 and not different (*Figure 2A and B*; p=0.27). With old mothers, the respective ratios were 1.10 vs. 1.1 and also not different (*Figure 2A and B*; p=0.62). Thus, as the new and old pole halves yield directly the new and old daughters upon the division of a mother, the placement of ribosomes in the mother cell sets up the quantity, and hence asymmetry, of ribosomes in the new and old daughters of a mother.

## Stochastic and deterministic components of ribosomal variance in daughter cells

Although new and old daughters can differ significantly in ribosome density, there was also substantial variation within each daughter population. A plot of the distributions shows clearly the within-population variation, although the difference between old and new daughters is also apparent (*Figure 3A and B*). Old and new daughters form their own distributions, but their mean values are displaced by a difference D, which is larger when the daughters are derived from old mothers. The variation within the old and new daughter distributions is presumably due to stochastic noise, although the stochasticity could also result from an unknown deterministic source. The latter is well illustrated by the discovery that *E. coli* mothers produce asymmetrical daughters. For example, without knowledge about the asymmetry, old and new daughters would have been pooled into one population with a total variance $V_T$ (*Figure 3A and B*; dashed lines), which would have been interpreted to result from stochasticity. In reality, $V_T$ would actually have a deterministic component due to the difference between old and new daughters. As we have shown for expressed gene products (*Shi et al., 2020*), $V_T = (V_{Old} + V_{New})/2 + D^2/4$, when there is deterministic asymmetry (D > 0) and $V_{Old}$ and $V_{New}$ are the variances of the unpooled old and new daughter populations (*Figure 3A and B*; red and blue lines). If D = 0, $V_T$ is the average of $V_{Old}$ and $V_{New}$. Thus, $D^2/4$ represents the deterministic component of $V_T$, and the contribution of deterministic asymmetry expressed as a percentage is $h^2 = (D^2/4)/V_T$. Our estimates of D showed that $h^2$ was 39.9% from old mothers and 2.4% from new mothers (*Table 1A*). The

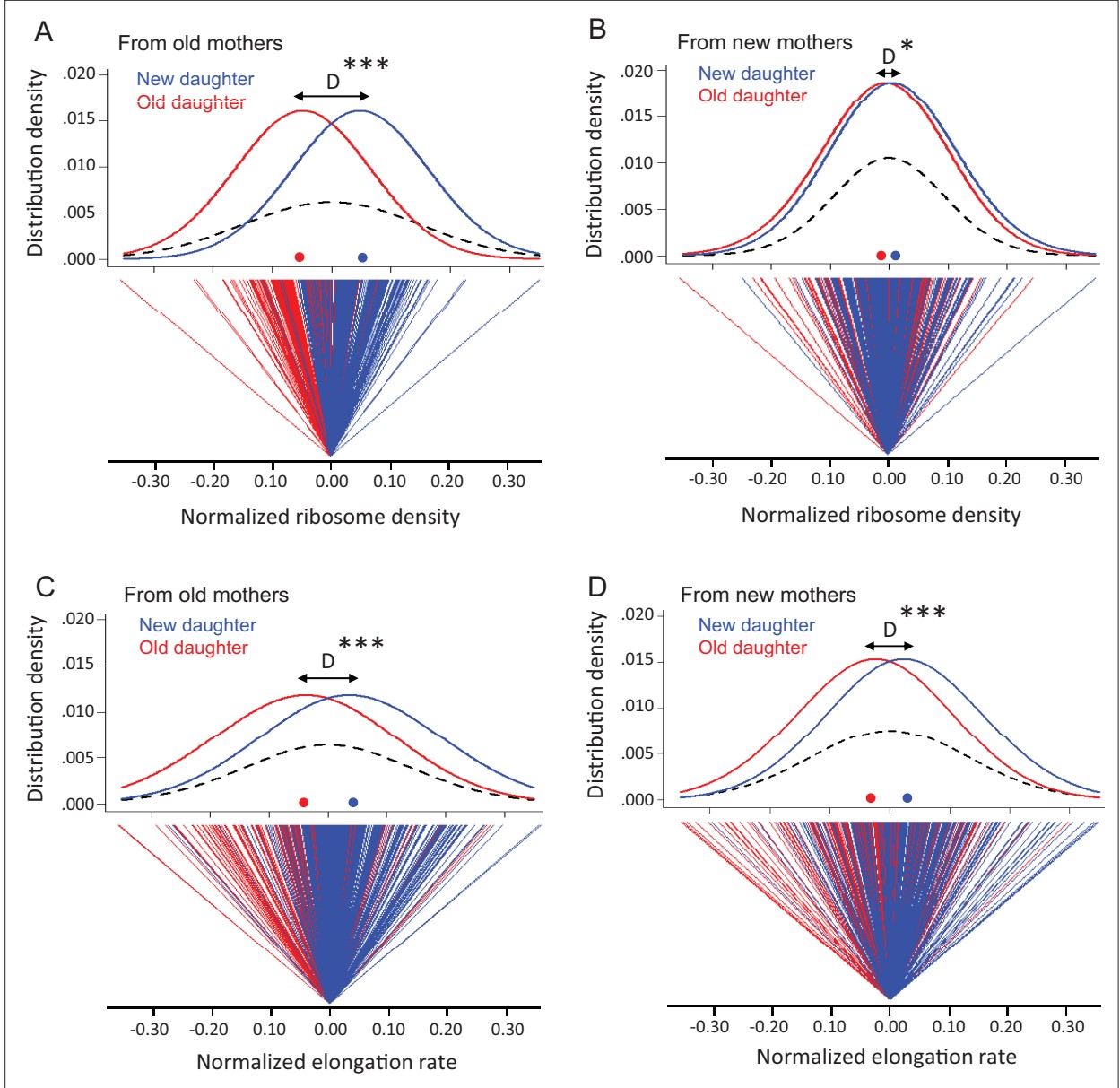

**Figure 3.** Variation in ribosome density and elongation rate in old and new daughters from old and new mothers. The measurements were made from the same cells in *Figure 2C*. All density and elongation values were normalized for this analysis. Means, variances, and other parameters extracted from these data are presented in *Table 1*. (**A**) Top panel: normalized density distribution of ribosome density of old (red line) and new (blue line) daughters from old mothers. Red and blue dots on the x-axis indicate the mean density for each distribution. Dashed lines represent the density distribution of the old and new daughters combined into one pooled total population. D (black arrow) shows the distance between peaks of old and new daughter curves (*** p=2.2 × 10$^{-16}$ for significance of D > 0; one-tailed paired *t*-test; n = 216 daughter pairs; see *Table 1A*). Bottom panel: normalized ribosome density of each old (red) and new (blue) daughter in pairs from old mothers. The zero point is set as the average ribosome density for each pair. As shown, the old daughter in each pair more often ends up on the minus side of the pair's zero point, i.e., having lower ribosome density. (**B**) Same as (**A**), but from new mothers (*p=0.014 for D > 0; one-tailed paired *t*-test; n = 198 daughter pairs; see *Table 1A*). (**C**) Normalized elongation rate distributions (***p=2.0 × 10$^{-11}$ for D > 0; one-tailed paired *t*-test; n = 216 daughter pairs; see *Table 1B*), but otherwise as (**A**). (**D**) Same as (**C**), but from new mothers (***p=2.1 × 10$^{-7}$ for D > 0; one-tailed paired *t*-test; n = 198 daughter pairs; see *Table 1B*).

higher h$^2$ for daughters from old mothers is consistent with our results in *Figure 3A and B* that show old mothers having more different daughters than new mothers. These results were also consistent with our previous report that h$^2$ is higher for expressed gene products in daughters of old than of new mothers (h$^2$ of 40.1 and 10.1%) (*Shi et al., 2020*). Thus, a claim of stochasticity to explain the variance of ribosomal density of *E. coli* cells could have introduced an error of nearly 40% without knowledge

**Table 1.** Variance components of ribosome density and elongation rates.

Total ($V_T$), stochastic ($V_E$), and deterministic ($D^2/4$) components of variances estimated from *Figure 3* (see 'Materials and methods'). The deterministic proportion of total variance $V_T$ was $h^2 = (D^2/4)/V_T$. The stochastic component of $V_T$ is therefore $V_E = V_T (1 - h^2)$. Significance testing for $D > 0$ as described in *Figure 3* (\*, \*\*, and \*\*\* denote $p<0.05$, $0.01$, and $0.001$). (A) Estimates for ribosomal density. (B) Estimates for elongation rates. (C) Comparison of $V_E$ estimates. Testing of significance difference between $V_E$ was done using Bartlett's test of homogeneity of variances. p-values[1] are for comparisons across rows (old vs. new mothers). p-values[2] are for significance of $V_E$ down columns (elongation rate vs. ribosome density).

**(A) Ribosomal density**

|  | Vr | D | D2/4 | h2 |
|---|---|---|---|---|
| Old mothers | 0.00634 | 0.101 *** | 0.00253 | 0.399 |
| New mothers | 0.00347 | 0.0184 * | 0.00008 | 0.024 |

**(B) Elongation rates**

|  | Vr | D | $D^2/4$ | $h^2$ |
|---|---|---|---|---|
| Old mothers | 0.00913 | 0.0821 *** | 0.00168 | 0.185 |
| New mothers | 0.00581 | 0.0533 *** | 0.00071 | 0.122 |

**(C) VE estimates**

|  | Old mothers | New mothers | p-value1 |
|---|---|---|---|
| Elongation rates | 0.00744 | 0.0051 | 0.0073** |
| Ribosome density | 0.00381 | 0.00338 | 0.40 n.s. |
| p-value[2] | 0.000001*** | 0.0041** |  |

of the asymmetry between old and new daughters. Without knowing more, we presume that the variances $V_{Old}$ and $V_{New}$ result from stochasticity, but it may also be that a deterministic process still lies hidden.

## Asymmetry and variance components for elongation rates in daughter cells

To determine whether the observed ribosome patterns reported in *Figure 3A and B* had a downstream effect, we measured elongation rates of the same daughters and mothers. When new and old daughters were pooled irrespective of whether they originated from old or new mothers, new daughters were found to elongate faster than old daughters (new/old ratio of $1.09 \pm 0.011$; $p=3.9 \times 10^{-16}$). When the old and new daughters were separated as descending from old or new mothers, the ratio was respectively $1.11 \pm 0.017$ ($p=1.3 \times 10^{-10}$) and $1.07 \pm 0.011$ ($p=2.8 \times 10^{-7}$) (*Figure 2C*). Although both ratios were significantly greater than 1.00, the asymmetry was greater between daughters from old mothers and the two ratios were significantly different ($p=0.02$). As we did for the ribosome data, the variance of elongation rate measurements was also portioned into a deterministic and stochastic component to estimate the percentage $h^2$ (*Figures 3C, D and 4B*). The deterministic percentage of the variance was found to be $h^2 = 18.5\%$ and $12.2\%$ for daughters originating from old and new mothers, respectively. As we observed for ribosome densities, the $h^2$ for old mothers was still higher, but not to the same magnitude (see 'Discussion'). The overall similarity between the ribosome and elongation rate data reflects the fact that the two phenotypes are well known to be positively correlated in *E. coli* (*Schaechter et al., 1958*; *Neidhardt and Magasanik, 1960*; *Kjeldgaard and Kurland, 1963*; *Poulsen et al., 1995*; *Greulich et al., 2015*). An examination of the relationship between our measurements for ribosome and elongation yielded similar outcomes (*Figure 4*). Daughters from both old and new mothers had ribosome densities that correlated positively with elongation rates ($r = 0.387$, $p<1 \times 10^{-5}$ and $r = 0.233$, $p=4 \times 10^{-4}$, respectively; see *Figure 4* for more details). The correlation from new mothers was weaker because its deterministic component $h^2$ was smaller (*Table 1B*), in which case the distributions of new and old daughters were less separated (*Figure 4A and B*).

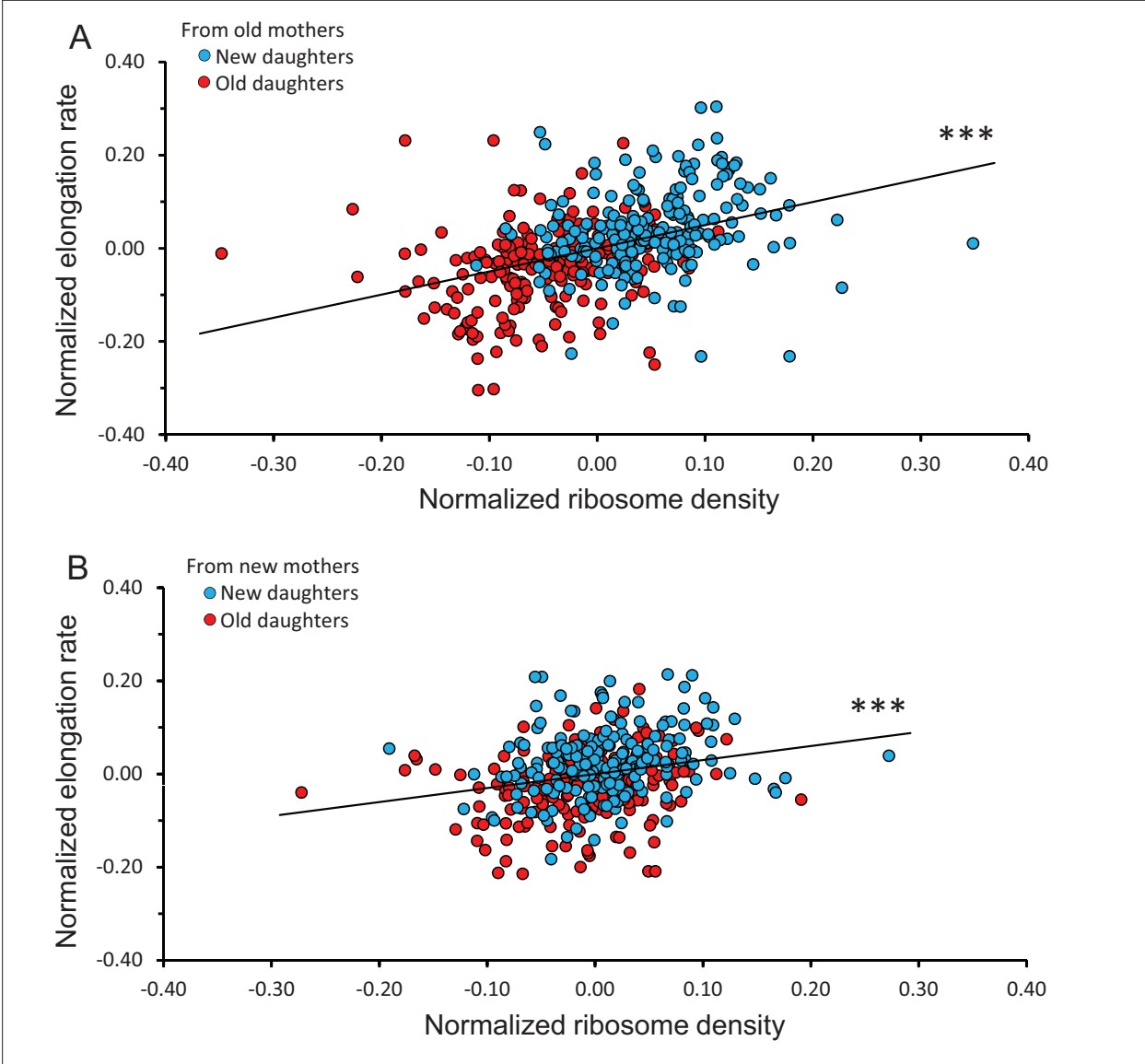

**Figure 4.** Correlation between normalized elongation rate and ribosome density of old and new daughters from old and new mothers. Because the old and new daughter values were jointly normalized, they were not independent and their combination into a single plot could not be assessed by statistical models assuming independence. Thus, all analyses for the figure were conducted by randomizing the data and obtaining a null distribution to estimate the significance values (see 'Materials and methods'). (**A**) Ribosome density versus elongation rate between new and old daughters from old mothers. Slope of linear regression = 0.498, correlation $r$ = 0.387, p<1 × $10^{-5}$, n = 216 daughter pairs. (**B**) Ribosome density versus elongation rate between new and old daughters from new mothers. Slope of linear regression = 0.301, correlation $r$ = 0.233, p=4 × $10^{-4}$, n = 198 daughter pairs.

## Spatial distribution of ribosomes in mother cells from birth to division

Our observation that the asymmetry of the pole halves of mothers determines the distribution of ribosomes of daughters at birth (*Figure 2A and B*) raises the question of how the spatial distribution of ribosomes changes temporally in the mother cell, and whether the changes provide insights or hypotheses for the origin of the asymmetry. Thus, we increased the resolution of the process by sampling over a finer spatial (lengthwise) and temporal (birth to division) grain of the ribosomal density in a mother cell. The length of the mother cell was divided into four quartiles, which were termed NP (new pole), L2, L3, and OP (old pole) (*Figure 1D*). The time from birth to division of the mother cell was also divided into four quartiles termed birth (B), T2, T3, and division (D). Ribosome density and the quartiles of length and time were all normalized for pooling the cells. Note that

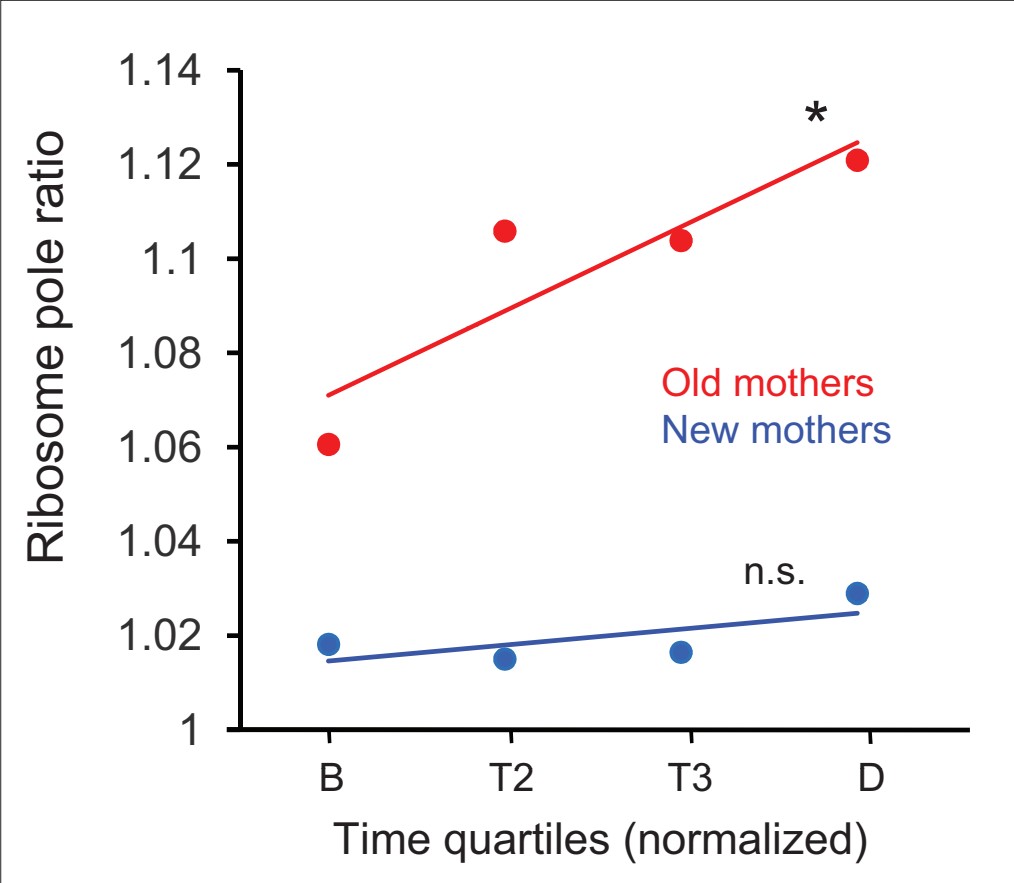

**Figure 5.** Ratio of ribosome density in old and new mothers over quartile time from birth to division. Ratios correspond to density in the new pole half divided by density in the old pole half (*Figure 1C*) of the mothers. Regression slopes were significantly greater than zero for old (slope = 0.0177, *p=0.026, df = 278) but not for new mothers (slope = 0.0039, n.s. p=0.59, df = 278). Regression was based on ratios that were binned by the four time quartiles (old mothers: n = 71, 74, 46, 89; df = 278; new mothers: n = 68, 72, 49, 91; df = 278), although the plots only show the ribosome means of the four time bins for illustration. Old mothers began with a higher asymmetry ratio and elevated to even higher levels at division. On the other hand, new mothers had a lower and more symmetrical ratio that was held more constant from birth to division.

because ribosome densities are normalized, they indicate relative rather than absolute amounts and the former can change even when the latter is constant.

We first examined whether the new length quartile results were consistent with our previous ribosome pole ratios (*Figure 2B*) for the new and old pole halves in the mothers. These new ribosome pole ratio data were obtained by taking the ratio of the length quartiles (NP +L2) / (L3 + OP) and plotting the values over the time quartiles (*Figure 5*). The ratios increased from birth to division in both old and new mothers, but increase was significant only for the old mothers (p=0.026 vs. 0.59, respectively). The rapid rise in the old mothers also began from a higher value at birth. More importantly, the final ratio values at quartile D (1.11 and 1.02 for old and new mothers, respectively) were comparable to *Figure 2C* (1.11 and 1.04), which was also timed to division. Thus, old mothers magnified the asymmetry of their ribosome distribution over time to achieve the higher ratio that we used above to explain the larger asymmetry of their daughters. On the other hand, the new mothers began with a pole ratio close to 1.02 and did not change over time, which explained the higher symmetry of their daughters.

By using the length into quartiles NP, L2, L3, and OP as individual measurements, we were able to examine in more detail the trajectories for the changes of ribosome densities of mothers from birth to division (*Figure 6A*). Several key patterns emerged. First, the trajectories decreased in NP and OP and increased in L2 and L3 in both old and new mothers. Moreover, the density at the final time

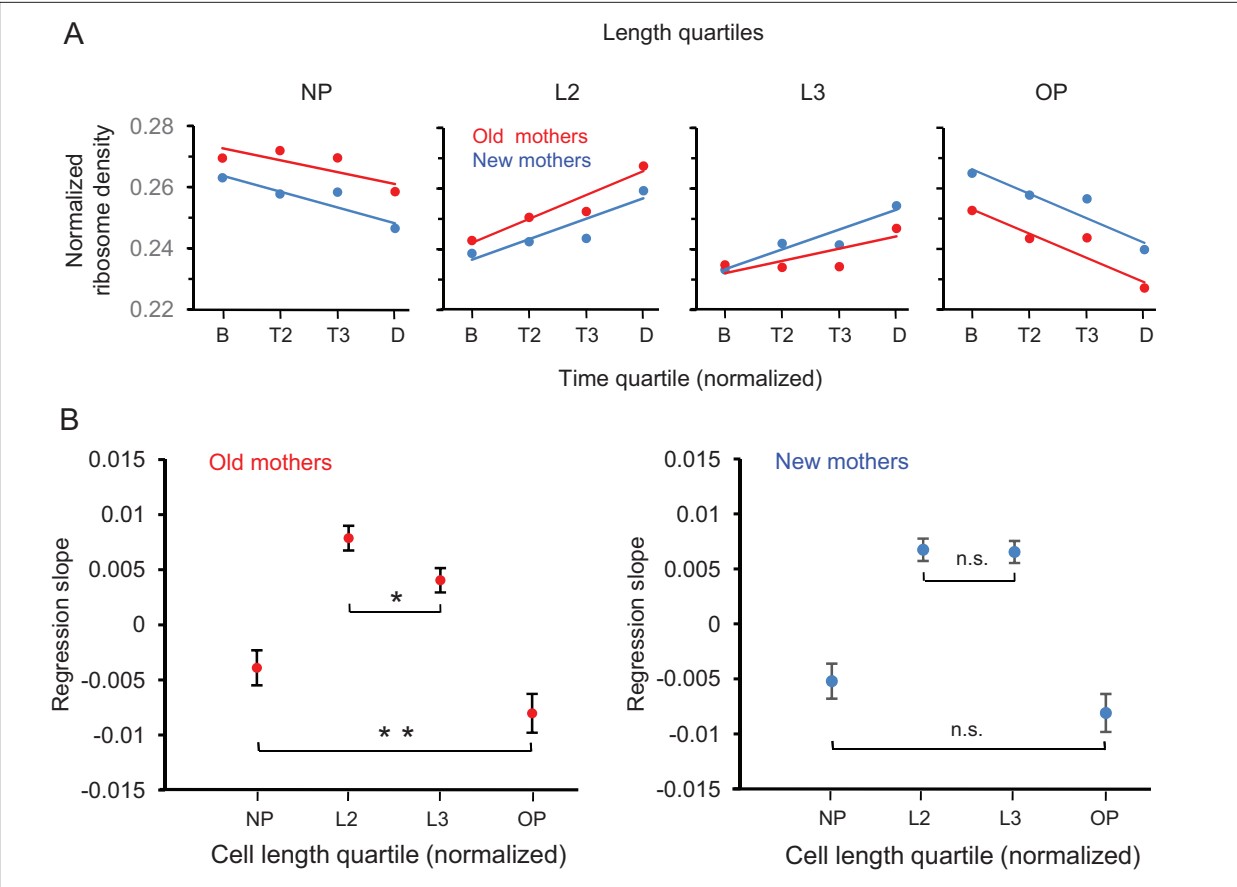

**Figure 6.** Spatial and temporal distribution of ribosome density in old and new mothers. (**A**) Density in length quartiles over time quartiles. Lines presented are from regressions of ribosome densities for one length quartile that were binned by the four time quartiles. All length quartiles for a mother type had time bins of the same sample size (old mothers: n = 71, 74, 46, 89; new mothers: n = 68, 72, 49, 91). The single dots presented are ribosome density means of the four time bins and are presented for illustration. (**B**) Values of regression slopes of the eight lines in (**A**). All regression slopes were significantly greater or less than, that is, not equal to, zero (old mothers: p=0.015, 1.46 × 10$^{-11}$, 0.00029, 7.8 × 10$^{-6}$; df = 278; new mothers: p=0.0012, 1.7 × 10$^{-10}$, 3.5 × 10$^{-10}$, 4.2 × 10$^{-6}$; df = 278). To test whether old mothers were more asymmetrical than new ones, the regression slopes of the pairs NP vs. OP and L2 vs. L3 for each mother type were compared (see 'Materials and methods' for details). The comparisons test whether length quartiles of the mothers on the left new pole side mirrors the right old pole side. The horizontal brackets in the figure denote the comparisons and whether they were significant. Old mothers were asymmetrical because their comparisons were all significantly different (NP vs. OP and L2 vs. L3; p=0.04 and 0.008; df = 158 and 112). New mothers were symmetrical as the differences were not significant (NP vs. OP and L2 vs. L3; p=0.11 and 0.44; df = 156 and 115).

quartile D was L2 > NP and L3 > OP. Because L2 and L3 are the new poles of the daughters after division (***Figure 1A*** and ***Figure 6A***), the daughters were already being provisioned by their mothers to be biased to have more ribosomes at their future new poles at the expense of their future old poles. Second, the trajectories of new mothers in NP-L2 were mirrored by L3-OP (***Figure 6A***; blue lines), that is, NP was more similar to OP, and L2 more similar to L3. The trajectory slopes of NP vs. OP (–0.00520 ± 0.00313 to –0.00809 ± 0.00340; p>0.05) and L2 vs. L3 (0.00675 ± 0.00200 vs. 0.00655 ± 0.00198; p>0.05) were not significantly different (***Figure 6B***). Thus, ribosomes were being redistributed from NP and OP to L2 and L3 in an equal manner and the lower asymmetry of new mothers was being maintained (***Figure 2B***). Third, the trajectories for the old mothers had a higher asymmetrical signature (***Figure 6B***; red lines). They resided above the new mother trajectories in NP and declined until they were below in OP. Moreover, the trajectory slopes of NP vs. OP (–0.00390 ± 0.00313 to –0.00803 ± 0.00347; p<0.05) and L2 vs. L3 (0.00788 ± 0.00220 vs. 0.00405 ± 0.00217; p<0.05) were now significantly different (***Figure 6B***). Thus, relative to L3 and OP, NP and L2 not only began at birth with more ribosomes, but also NP lost fewer and L2 acquired more ribosomes over time. As a result, the old mothers became even more asymmetric than new mothers at division.

# Discussion

The discovery that the division of mother *E. coli* cells produces two asymmetric daughters that have different elongation rates was groundbreaking (*Stewart et al., 2005*). The old daughter receiving the maternal old pole (*Figure 1A*) elongates more slowly than the new daughter receiving the maternal new pole. The difference is not due to genetic mutations because the daughter with the slower rate can in turn produce a grand-daughter with the elevated rate (*Stewart et al., 2005*; *Chao, 2010*; *Proenca et al., 2018*; *Proenca et al., 2019*; *Rang et al., 2011*). These results have been replicated on agar surfaces (*Stewart et al., 2005*; *Shi et al., 2020*; *Rang et al., 2018*; *Rang et al., 2012*) and in liquid media (*Proenca et al., 2018*; *Proenca et al., 2019*; *Łapińska et al., 2019*; *Rang et al., 2011*), although the tracking of cells by microscopy requires the cells to be a monolayer in both methods. Population growth rates during microscopy are comparable to those in shaking liquid cultures (*Proenca et al., 2018*; *Proenca et al., 2019*; *Łapińska et al., 2019*; *Lindner et al., 2008*; *Wang et al., 2010*). The causal trigger for the asymmetry between the daughters remains unknown, although damaged proteins (*Lindner et al., 2008*), expressed gene products (GFP), and now ribosomes (*Figure 5*; see below) have been shown to correlate with the differences in elongation rate. Damage may have the strongest causal link because the addition of external damage agents such as heat, phototoxicity, and antibiotics increases differentially the death probability of old daughters (*Proenca et al., 2018*; *Proenca et al., 2019*). Because new daughters are born with less damage, they are not killed by the damage agents and allow for the survival of the population (*Chao, 2010*; *Proenca et al., 2019*). The elongation rate differences have an epigenetic effect because slow mothers produce slower daughter pairs, although the new daughter is still quicker within the pair (*Chao, 2010*; *Proenca et al., 2018*; *Proenca et al., 2019*; *Rang et al., 2011*). If the causal factor were damaged proteins, the epigenetic interpretation is that the slower mother has more damage than average, and therefore the old and new daughters receive also more than their respective averages but the asymmetry is preserved because the former still gets more than the latter.

Because old daughters are more vulnerable to cell death than new daughters (see above), the elongation rate asymmetry can be considered a form of biological aging (*Stewart et al., 2005*). Thus, *E. coli* is a model for the evolutionary origin of biological aging from single-celled organisms. We would extend the argument to suggest that the old daughters are more properly considered to be the continuation of mother, and the new daughters are the true daughters. Given an epigenetic factor that slows the elongation rate of a cell, the mother retains a larger pool during division and it ages. The daughter is allocated less and it is rejuvenated. The larger pool that the mother does not transmit to the daughter is the original soma. Aging is the price that a mother pays to produce healthy daughters. Because the elongation rate asymmetry between mothers and daughters maximizes the fitness of the lineage (see 'Introduction'), aging is indirectly favored by natural selection (*Chao, 2010*; *Chao et al., 2016*). Because the origin of life required organization of genetic material into protocells (*Chao, 1991*) in hazardous damage-inducing conditions (*Miller and Urey, 1959*; *Michaelian, 2011*), biological aging may have evolved in the first life forms.

The higher vulnerability of old daughters (or old mothers) in the presence of added damage agents was quantified by estimating the mean $a$ and the variance $s^2$ of the regression (or mapping) coefficient of the doubling time (DT) of daughters onto mothers (*Proenca et al., 2018*). The vulnerability is high if $(a^2 + s^2) < 1$ because the daughter could then acquire a very long DT and fail to divide. For example, if $a = 1.1$, a mother with DT = 40 min would have a daughter with a longer DT = 1.1 * 40 min, and the DT of each subsequent daughter in the lineage is even longer. If $s^2$ is large, then by chance the DT of a daughter could also be vulnerably long. When damage agents were added, the old daughters were the first ones to die because their values of $a$ and $s^2$ increased above the threshold (*Proenca et al., 2019*). Thus, the asymmetry between old and new daughters is extended to include the values of $a$ and $s^2$.

A question that is often raised is why evolution by natural selection is not able to minimize $a$ and $s^2$ to diminish the window of vulnerability. An answer is that it does, but the damage agents are pathological conditions beyond the physiological limits. We suggest an additional answer for $s^2$ based on our variance measurements for ribosome density and elongation rate (*Table 1*). From those measurements, we estimated the stochastic and deterministic components of the variances. The deterministic component, which resulted from the old and new daughter asymmetry, was $D^2/4$. The stochastic component is therefore $V_E = V_T - D^2/4 = V_T (1 - h^2)$. $V_E$ is comparable to $s^2$ because both

describe phenotypic variation in a clonal population and are generated by stochastic noise. From our measured values (*Table 1C*), $V_E$ for ribosome density was 0.0038 and 0.0034 in old and new mothers and not significantly different (p=0.40). For elongation rate, $V_E$ was 0.074 and 0.051, significantly higher in old mothers (p=.007), and comparable with the trend reported for $s^2$ (*Proenca et al., 2019*). A cross-comparison of $V_E$ for ribosome and elongation showed that the latter was significantly higher both in old mothers (0.00381 vs. 0.00744; p=1 × 10$^{-6}$) and in new mothers (0.00338 vs. 0.00510; p=0.0041). This inflation of the elongation $V_E$ is consistent with the hypothesis that downstream traits are inherently noisier because they are at the end of many converging pathways or networks, each one with its own inherent noise (*Price and Schluter, 1991*; *Houle, 1992*). Because ribosomes are farther upstream, they have fewer inputs and a lower final $V_E$. Thus, another answer is that although complex traits such as elongation are important for a cell and under strong natural selection, they are most likely downstream and their $V_E$ are noisier and higher in value.

Previous studies have shown that old and new daughters in *E. coli* are asymmetrical for elongation rates (*Stewart et al., 2005*; *Proenca et al., 2018*; *Proenca et al., 2019*; *Łapińska et al., 2019*; *Lindner et al., 2008*; *Chao et al., 2016*; *Rang et al., 2011*), gene expression (*Shi et al., 2020*), and vulnerability to dying in the presence of damage agents (*Proenca et al., 2019*). We show here that the variances ($V_E$) of ribosome density and elongation rates are also asymmetric. If asymmetry is a central characteristic of bacterial aging, identifying its cause becomes an important goal. We began our studies by examining asymmetry for elongation rates, and then to gene products and now to ribosomes. Because ribosomes are positioned highly upstream in the pathway of these cells and their abundance correlates strongly with elongation rates in *E. coli* (*Schaechter et al., 1958*; *Neidhardt and Magasanik, 1960*; *Kjeldgaard and Kurland, 1963*; *Poulsen et al., 1995*), their physiological role is both fundamental and general. As we have argued, our previous report on the asymmetry of gene products was instructive but left uncertain whether the outcome was unique to the gene that was studied or a more general phenomenon. Our present results on ribosome asymmetry move the process to a more fundamental level because ribosomes can affect more genes and more downstream processes. More importantly, existing measurements of ribosome cytology and cell growth in *E. coli*, in conjunction with our new results, allow us to quantitatively evaluate, understandingly not to prove, our previous suggestion (*Shi et al., 2020*) that space competition between damage aggregates and ribosomes could be responsible for the asymmetry.

The nucleoid and ribosomes in a rod-shaped *E. coli* occupy non-overlapping space along the length of the cell (*Bakshi et al., 2012*). At birth, the cells are smaller and the nucleiod resides at the mid-region of the rod and the ribosomes are displaced to the poles. As the cell elongates, the nucleoid splits to create the two nucleoids for the daughters. The splitting opens space in mid-region and the void is then filled by the ribosomes. Cell division ensues with a cleavage at the mid-region, and the ribosomes at the mid-region are distributed to the new poles of the daughters. However, the data used to generate this description did not distinguish between old and new poles or daughters. As a result, the cells were oriented randomly, and the resulting profiles are generic averages and necessarily symmetrical. When we distinguished between the old and new poles and daughters, the ribosome distribution resembles the generic profile only in new mothers and is asymmetrical in old mothers (*Figure 6*). Because damage aggregates also tend to reside at the poles (*Lindner et al., 2008*), we need to evaluate whether the volume of added aggregates displaced a volume of ribosomes that could quantitatively account for the elongation rate difference between old and new daughters from old mothers (*Figure 2C*). Such an analysis is possible using published information.

Single-cell measurements estimate elongation rates for *E. coli* to be 2.148 and 1.964 hr$^{-1}$ for new and old daughters in rich media (*Proenca et al., 2018*). The well-known linear relationship between elongation rate and ribosomal content (*Schaechter et al., 1958*; *Neidhardt and Magasanik, 1960*; *Kjeldgaard and Kurland, 1963*; *Greulich et al., 2015*) can be quantified from the data (*Poulsen et al., 1995*) as Y = 337.046 · X+87.167, where Y is ribosomal content (A.U.) and X is elongation rate (hr$^{-1}$). The relationship between ribosome density and elongation rate in *Figure 4* could not be used because the values were normalized. Letting $X_{New}$ = 2.148 and $X_{Old}$ = 1.964 hr$^{-1}$ from above, $Y_{New}$ = 811.02 and $Y_{Old}$ = 749.08 A.U. The proportional reduction of ribosomes in the old daughter relative to the new daughter is therefore $p$ = (811.02–749.08)/811.02 = 0.0764. The total volume occupied by ribosomes in rich media is estimated to be $V_{ribo}$ = 0.79 µm$^3$ (*Woldringh and Nanninga, 1985*). The volume needed to reduce the elongate rate of the old daughter to 1.964 hr$^{-1}$ becomes $\Delta = p \cdot V_{ribo}$

= 0.0764 · 0.79 = 0.0603 µm$^3$. Aggregate radii have been estimated from diffusion constants for *E. coli* cells in rich media (*Coquel et al., 2013*). By converting the radii to a sphere as $(4\pi/3) \cdot \text{radius}^3$, the largest aggregates had mean values of 0.0852 µm$^3$, which is well in the range of our estimate of $\Delta = 0.0603$ µm$^3$. Thus, space competition between aggregates and ribosomes could quantitatively account for the lower elongation rate of old daughters of old mothers.

Our above space competition model implies that elongation rates and the size of damage aggregates should be negatively correlated. However, while the initial study by *Lindner et al., 2008* found a strong and significant negative correlation (p<0.0001), a later study by *Govers et al., 2018* found one that was negative but nonsignificant (p=0.08). Lindner et al. and Govers et al., respectively, used different stressors to induce damage (streptomycin and heat shock) and fluorescent reporters (YFP and msfGFP) to quantify aggregate size. The YFP and msfGFP used in both studies were fused to the chaperone IbpA, which associates to aggregates of damaged proteins. Because Govers et al. believed that IbpA-YFP formed self-aggregates they suggested that the correlation of Lindner et al. was an artifact of YFP self-aggregation. However, the rejection of Lindner et al. may be premature. First, the suggestion of YFP self-aggregation by *Govers et al., 2018* was based on their demonstration that IbpA-YFP increased the expression of IbpA (at another locus), which is induced by perturbations to protein homeostasis. The expression was not increased in the IbpA-msfGFP control. Because the assay was conducted in the absence of heat shock or another stressor, they surmised that it was YFP self-aggregates that had perturbed the homeostasis. However, a recent report that IbpA self-represses its own translation because over-expression is harmful in the absence of stress (*Miwa et al., 2021*) suggests an alternative explanation for why IbpA-YFP increased expression. It could be that IbpA-YFP was unable to self-repress because of its larger size. Second, msfGFP-YFP may have its own issues because super-folding GFPs are more prone to photobleaching (*Valbuena et al., 2020*; *Pédelacq et al., 2006*). Could photobleaching have weakened the correlation of Govers et al. and rendered it nonsignificant? Third, most importantly, the correlations of Lindner et al. and Govers et al. must be compared with caution. While Lindner et al. induced cell damage with 10 µg ml$^{-1}$ of streptomycin, where the minimal inhibitory concentration is 16–32 µg ml$^{-1}$ (*Tashiro et al., 2017*; *Regoes et al., 2004*), Govers et al. used 47°C heat shock for 15 min, where 50°C at 15 min is lethal (*Govers et al., 2018*). Although both Lindner et al. and Govers et al. grew their cells in LB broth, the elongation rate in the two studies, respectively, centered about 0.35 hr$^{-1}$ and 0.25 hr$^{-1}$ (19 and 27 min doubling times). To have a doubling time of 27 min in LB, the cells in Govers et al. must have experienced severe stress and damage. In relative terms, the two treatments, respectively, were 10/16 to 10/32 = 63% to 31% vs. 47/50 = 94% of stopping growth. It is likely that at 94% lethality the stress level in Govers et al. is collapsing most cellular processes, including protein aggregation. This possibility is supported by Govers et al.'s measurement of the mean IbpA-msfGFP fluorescence per cell as a function of increasing heat shock temperature. The mean response increased linearly from 42°C up to 48°C, and then from 48°C to 50°C it crashed abruptly back to the control level (shock-free) (*Govers et al., 2018*). If the accuracy of laboratory incubators is ± 1°C, Govers et al. were using IbpA-msfGFP dangerously near the failure point of the reporter, which could have underestimated (or estimated with more error) the true amount of protein damage in the cells. If a 47°C heat shock induced damage to other cellular components besides proteins (e.g., DNA), the IbpA-msfGFP would not have reported them. High heat can damage DNA and elevate mutation rates in *E. coli* (*Zamenhof and Greer, 1958*; *Chu et al., 2018*). The choice of 47°C was justified because Govers et al. wanted to test whether protein aggregates could protect cells from new damage. However, if the underestimates or statistical analyses weakened by measurement errors compromised the correlation of Govers et al., the two correlations are not equivalent and cannot be used to reject each other.

Until new studies prove otherwise, we believe that the elongation rate and aggregate correlations of Lindner et al. and Govers et al. should be viewed as alternative models to be further tested. Although the correlations do not bear directly on our ribosome results, they are relevant to our discussion of the relationship of ribosomes, damage aggregates, and elongation rates. Our space competition model (see above) was evaluated to test whether our data was (or was not) quantitatively consistent with the correlation reported by Lindner et al. We look forward to other tests or challenges as connecting the two correlations will be critical. The protection uncovered by Govers et al. is an exciting new phenomenon. However, we are most excited by the possibility that both correlations are correct and protein aggregates are harmful and protective under low and high stress, respectively. At low stress levels,

proteins could be the main target of damage and the amount of protein aggregates correlates with elongation rate. The damaged proteins are aggregated and then partitioned asymmetrically between the daughters. Protein repair is occurring (*Ezraty et al., 2017*; *Merdanovic et al., 2011*) and its effect is to lower the damage rate experienced by the cell. If aggregates are observed, damage rates must be exceeding repair rates. However, a full stress response (e.g., SOS, efflux transporters, and RpoS) (*Roemhild et al., 2022*) is too costly and unnecessary. At high stress levels and near lethality, proteins, DNA, and other cellular processes (including protein aggregation and IbpA-msfGFP reporting) are all failing due to damage and the correlation between elongation rate and protein aggregates is weakened. In order to survive, the cell needs a full stress response and aggregates become beneficial because they are present to act as a trigger (or the 'epigenetic memory' of Govers et al.). In effect, protein aggregates at high levels are inducing persister states (*Leszczynska et al., 2013*; *Dewachter et al., 2021*; *Goode et al., 2021*; *Balaban et al., 2019*).

In summary, previous reports have documented in *E. coli* the asymmetrical associations between bacterial aging, elongation rates, cell poles, damage aggregates, and expressed gene products, and we have added one more phenotype, the density of ribosomes. Because ribosomes have a fundamental role in determining cell function, our results indicate that bacterial aging and the asymmetry are resulting from processes that are central to cell function and not just a subset of genes or pathways. Moreover, ribosomal function has been well characterized and the information allowed us to show that space competition between ribosomes and damage aggregates, both of which tend to reside in the poles, could quantitatively account for the elongation rate asymmetry between old and new daughters. The latter calculation does not prove a causal role for damage aggregates, but we hope it will motivate and guide new research to evaluate hypotheses emanating from the work by us and others.

## Materials and methods
### Bacterial strain
*E. coli* AFS55 was used to quantify ribosome density. AFS55 has a translational fusion of *yfp* (yellow fluorescent protein) to the C-terminus of *rpsB* (ribosomal subunit S2) that served as a reporter for ribosome density (*Bakshi et al., 2012*). AFS55 was chosen for this study because it offers currently one of the best fluorescent reporters for ribosomes (see 'Introduction' for full details). For example, numerous controls have been conducted to rule out non-uniform fluorescent distribution by YFP self-aggregation artifacts (*Bakshi et al., 2012*; *Batchelor and Goulian, 2006*). Additionally, the S2 protein was picked from other possible ribosomal proteins because it is known to have a negligible number of free copies (not attached to ribosomes) in the cell (*Bakshi et al., 2012*; *Sanamrad et al., 2014*).

### Growing cells and preparing microscopy slides
All cells were stored in 40% glycerol at –80°C and incubated at 37°C. A glycerol stock was streaked onto LB agar plates (*Sambrook et al., 1989*) to obtain single colonies. Liquid cultures were made by inoculating a single colony into 10 ml M9 (*Sambrook et al., 1989*) media and incubating overnight. The overnight culture was diluted 1:100 in M9 and grown for two more hours. 1 µl of 2-hr culture was then pipetted onto a 10 µl agarose pad with M9 and 15% agarose. The agarose pad was flipped with the bacterial side down onto a 24 × 60 mm cover glass, which was then placed over a 25 × 75 mm single depression slide. The contact between the cover glass and slide was sealed with Vaseline to prevent desiccation. These methods were modified from *Rang et al., 2011*; *Rang et al., 2012* and *Stewart et al., 2005*.

### Microscopy and time-lapse movies
After cells were placed onto agar pads, the microscopy slides imaged at 37°C with an inverted microscope (Nikon Eclipse Ti-S), equipped with Nikon NIS-Elements AR control software, ×100 objective (CFI Plan APO NA 1.4), external phase-contrast rings for full-intensity fluorescence imaging, fluorescence light source (Prior Lumen 200) with motorized shutter (Lambda 10-B Sutter SmartShutter), and camera (Retiga 2000R FAST 1394, mono, 12 bit). To be able to quantify the fluorescence of growing single cells, movies were started and ended following guidelines. To prevent crowing as cells grew into micro-colonies over time, movies were started from microscope fields containing single cells with

distant neighbors. To avoid the effects of crowding within the micro-colonies, movies were ended when micro-colonies exceeded 128 cells. Phase-contrast and fluorescence images were recorded every 2 and 20 min, respectively. Thus, every tenth phase image has a matching fluorescence image taken at the same time and with the same X–Y coordinates. Such fluorescence×phase pairs were used for our deconvolution protocols (see the next section).

Cells to be measured were collected from colonies started by a single cell. Measurements were made at generations 4, 5, 6, and 7, which correspond to colonies with 16, 32, 64, and 128 cells. Smaller sizes (1, 2, 4, and 8 cells) were not measured because a colony started from a single cell experiences a lag, despite being inoculated with fresh exponentially cells. Colony sizes larger than 128 were also excluded because cell growth slows down due to crowding within and between colonies. Thus, to be safe and ensure the most uniform physiological state for the cells that were measured, we restricted our measurements to colonies with 16–128 cells.

## Image processing, discounting light scatter from neighbors, outlining cells, and deconvolution

Fluorescence measurements reporting ribosome density in single cells were obtained from the fluorescence images. Because a cell in a micro-colony receives scattered fluorescence light from neighboring cells, the value of a fluorescence pixel inside a cell is the sum of the true emission of the pixel plus the scatter. Note that background pixels, which are outside a cell and lacking a fluorescent reporter, should have a true emission of zero (not be emitting) and only contain scatter from pixels in the neighboring cells. The true emission can be estimated by dividing the image into all its pixels, treating the pixels as light point sources, using a system of differential equations to describe the scatter from every pixel to every other pixel, and solving the system. Because every fluorescence image has a matching phase image (see above fluorescence–phase pair), any single cell can be outlined in the phase image. The correct total amount of fluorescence in the cell is therefore the true emission of all pixels contained by the outline. However, because in practice the value of a pixel is measured with error (stochasticity), the system cannot be solved analytically and must be solved numerically by the iterative process of deconvolution (*Lucy, 1974*). Deconvolution introduces a new problem because the number of iterations needs to be optimized. The true emission is underestimated if too few iterations are conducted (*Biggs and Andrews, 1997*). Too many iterations also underestimate, but by introducing ringing (*Mosleh et al., 2014*). Ringing results when a background pixel is brighter than average and is treated as an emitting pixel by the over-iteration. As a result, the over-iteration removes light from the true emission value of the pixels inside cells and allocates them to such background pixels. By tracking and plotting the sum of all pixels in a cell as a function of iterations, we have found that the sum invariably produced a peak by increasing initially and then decreasing (*Shi et al., 2020*). The number of iterations corresponding to the peak is chosen as the desired number that balanced the opposing effects of under and over-iterating. Because a detailed description has been published in *Shi et al., 2020*, we present next a more abbreviated technical summary.

The background of the fluorescence–phase image pairs (see above) was first subtracted using the 'rolling ball' algorithm in ImageJ (NIH) with ball radius 20 pixels. The noise created by the heat overflow of single pixel was corrected using the 'remove outliers' algorithm in ImageJ with a threshold intensity difference of 1000 and threshold radius of 0.5 pixel. The resulting image was processed by iterative Lucy–Richardson deconvolution in MATLAB 2017b using an optimal number of iterations for each fluorescence–phase pair as described above. The outline of a single cell was traced manually with ImageJ. Blind replicates without any awareness of the history of the cells found no bias between outlines measured for old and new mothers.

## Statistical analyses, data measurements, and normalization

Standard *t*-test and regression analyses were implemented with packaged R functions. All reported error bars are standard error of the mean (SEM).

Following deconvolution, fluorescence images and the outlines of cells were imported as data matrices into computational R-programs. For fluorescence measurements reporting ribosomal density, the outlines were cut into the desired number of partitions (halves or quartiles; see *Figure 1*), the partitions were superimposed onto the fluorescence images, and the fluorescence pixel values contained by each partition were summed. The sum was then divided by the area of the partitions

to obtain the average pixel value or density. The elongation rate of a cell was estimated as $r = \ln(L_T/L_0)/T$, where $L_T$ and $L_0$ are the lengths of the cell when it was born and divided, respectively, and T is the time to division (min). Cell length measured manually from the phase images with ImageJ. A log transformation was used because elongation rates are known to be exponential (*Stewart et al., 2005*).

For *Figure 3A and B*, ribosome densities (here denoted X) were normalized as $(X – u)/u$, where $u = (X_{Old} + X_{New})/2$. For *Figure 3C and D*, elongation rates were likewise normalized. For *Figure 6*, the cells were pooled by first dividing each cell into four length quartiles (NP, L2, L3, and OP; see *Figure 1D*). Each cell was then normalized by dividing the ribosome density of each of the cell's quartiles by the sum of the ribosome density of the cell's four quartiles. After normalization, the sum equals 1.0. For statistical comparisons between two regression slopes (*Figure 6B*), the SEM values and sample sizes from the regression were used to estimate the resulting *t*-statistic and ensuing p-value (*Sokal and Rohlf, 1995*).

To pool cells for comparison over a time cycle from birth to division, time was normalized into quartiles (B, T2, T3, and D; birth, quartile 2, quartile 3, and division). For example, given a cell with a division time of 28 min and time-lapse images of the cell from birth to division, the images from the window $t = [0–7]$, $[7–14]$, $[14–21]$, and $[21–28]$ min. Because time quartiles are dimensionless, we provide as a reference the mean division times that were observed. The means for old and new daughters (depicted in *Figure 3C and D*) were, respectively, 74.58 and 67.48 min (from old mothers; n = 216) and 69.76 and 65.71 min (from new mothers; n = 198).

All statistical analyses were conducted with both length and ribosome densities values normalized as halves or quartiles. We provide in *Figure 1—figure supplement 1* a plot of ribosomal density versus length as a continuous variable for visualization and reference.

## Acknowledgements

We thank James Weisshaar for the bacterial strain. This study was supported by the NSF grant (DEB 1354253) to LC.

## Additional information

### Funding

| Funder | Grant reference number | Author |
| --- | --- | --- |
| National Science Foundation | DEB-1354253 | Lin Chao |

The funders had no role in study design, data collection and interpretation, or the decision to submit the work for publication.

### Author contributions

Lin Chao, Conceptualization, Resources, Data curation, Software, Formal analysis, Supervision, Funding acquisition, Validation, Investigation, Visualization, Methodology, Writing – original draft, Project administration, Writing – review and editing; Chun Kuen Chan, Software, Formal analysis, Validation, Investigation, Visualization, Methodology, Writing – original draft, Project administration, Writing – review and editing; Chao Shi, Software, Formal analysis, Supervision, Validation, Investigation, Visualization, Methodology, Project administration, Writing – review and editing; Ulla Camilla Rang, Conceptualization, Resources, Software, Supervision, Validation, Investigation, Visualization, Methodology, Writing – original draft, Project administration, Writing – review and editing

### Author ORCIDs

Lin Chao https://orcid.org/0009-0005-4114-5841
Ulla Camilla Rang https://orcid.org/0000-0002-3589-7327

Reviewer #1 (Public Review): https://doi.org/10.7554/eLife.89543.3.sa1
Author response: https://doi.org/10.7554/eLife.89543.3.sa2

## Additional files

### Supplementary files

• MDAR checklist

### Data availability

All data analyzed and presented in Figures are available in Dryad in https://doi.org/10.5061/dryad.59zw3r2jg.

The following dataset was generated:

| Author(s) | Year | Dataset title | Dataset URL | Database and Identifier |
|---|---|---|---|---|
| Chao L, Chan CK, Shi C, Rang CU | 2024 | Spatial and temporal distribution of ribosomes in single cells reveals aging differences between old and new daughters of *Escherichia coli* | http://dx.doi.org/10.5061/dryad.59zw3r2jg | Dryad Digital Repository, 10.5061/dryad.59zw3r2jg |

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
