## [Editor Report · eLife assessment]

This study is a potentially **important** contribution to the field of protein biosynthesis pathways and their link to aging, especially regarding the thorough analysis of variation in measures expected to correlate with elongation rate in old and new daughter cells derived from old and new mother cells. However, the imaging results, analysis, and methodologies are **incomplete**, as in its current form several key questions remain unanswered.

---

## [Referee Report · Reviewer #1 (Public Review)]

The research by Lin Chao, Chun Kuen Chen, Chao Shi, and Camilla U. Rang addresses the asymmetric distribution of ribosomes in single *E. coli* cells during aging by time-lapse microscopy, as well as its correlation to protein misfolding. The presented research is an important contribution to the field of protein biosynthesis pathways and their link to aging, especially in regard to the thorough analysis of variation in cells elongation rate in old and new daughter cells derived from old and new mother cells.

Comments on current version:

I thank the authors for their thoughtful responses. Yet the centrality of protein aggregate distribution analysis to this manuscript requires further evidence to support the link to ribosome asymmetrical distribution and aging.

The authors suggest this is beyond the scope of this study. This then requires a major revising of the study, as in its current form, it is one of its main claims.

---

## [Author Response]

The following is the authors’ response to the original reviews.

Response to reviewers

A general comment was that this study left several key questions unanswered, in particular the causal mechanism for the reported ribosomal distributions. We have been interested in the evolution of asymmetric bacterial growth and aging for many years. However, a motivational difference is that we are more interested in the evolutionary process, and evolution by natural selection works on the phenotype. Thus, we wanted to start with the phenotype closest to fitness, appropriately defined for the conditions, work downwards. We examined first the asymmetry of elongation rates in single cells, then gene products, and now ribosomes. As we have pointed out, our demonstration of ribosomal asymmetry shows that the phenomenon was not peculiar and unique to the gene products we examined. Rather, the asymmetry is acting higher up in the metabolic network and likely affecting all genes. We find such conceptual guidance to be important. In the ideal world, of course we would have liked to have worked out the causal mechanisms in one swoop. In a less than ideal situation, it is a subjective decision as where to stop. We believe that the publication of this manuscript is more than appropriate at this juncture. We work at the interface of evolutionary theory and microbiology. Our results could appeal to both fields. If we attract new researchers, progress could be accelerated. Could the delay caused by publishing only completed stories slow the rate of discovery? These questions are likely as old as science (e.g., https://telliamedrevisited.wordpress.com/2021/01/28/how-not-to-write-a-response-to-reviewers/).

We present below our response to specific comments by reviewers. We have not added a new discussion of papers suggested by Reviewer #1 because we feel that the speculations would have been too unfocused. We were already criticized for speculation in the Discussion about a link between aggregate size and ribosomal density.

Respond to Major comments by Reviewer #1.

a) Fig. 1 only shows 2 divisions (rather than 3 as per Rev1) to avoid an overly elaborate figure. We have added text to the figure legend that the old and new poles and daughters in the subsequent 3, 4, 5, 6, and 7 generations can be determined by following the same notations and tracking we presented for generations 1 and 2 in Fig. 1. For example, if we know the old and new poles of any of the four daughters after 2 divisions (as in Fig. 1), and allow that daughter to elongate, become a mother, and divide to produce 2 “grand-daughters”, the polarity of the grand-daughters can also be determined.

b) Because division times were normalized and analyzed as quartiles, the raw values were never used. Rather than annotating unused values, we have provided the mean division times in the Material and Methods section on normalization to provide representative values.

c) We did not quantify in our study the changes over generations for three reasons. First, the sample sizes for the first generations (cohorts of 1, 2, 4, and 8 cells) are statistically small. Second, and most importantly, cells on an agar pad in a microscope slide, despite being inoculated as fresh exponentially growing cells, experience a growth lag, as all cells transferred to a new physiological condition. Thus, to be safe, we do not collect data from cohorts 1, 2, 4, and 8 to ensure that our cells are as much as possible physiologically uniform. Lastly, as we noted in the Material and Methods they also slow down after 7 generations (128 cells). Thus, we have collected ribosome and length measurements primarily from cohorts 16, 32, 64, and 128. Measurable cells from the 128 cohort are actually rare because a colony with that many cells often starts to form double layers, which are not measurable. Most of our measurements came from the 16, 32, and 64 cohorts, in which case a time series would not be meaningful. Some of these details were not included in our manuscript but have been added to the Material and Methods (Microscopy and time-lapse movies). For these reasons we have not added a time series as requested by the reviewer.

d) We have added the additional figure as requested, but as a supplement rather than in the main article (Supplemental Materials Fig. S1). This figure showed the normalized density of ribosomes along the normalized length of old and new daughters. The density was continuous rather than quartiles. This figure was included in the original manuscript, but readers recommended that it be removed because the all the analyzed data had been done with quartiles. Readers felt mislead and confused.